# Methodological proposal to identify the nationality of Twitter users through random-forests

**Damián Quijano**[1,2‡]*, **Richard Gil-Herrera**[3‡]

**1** Department of Engineering and Computer Sciences, Universidad Especializada de las Américas (UDELAS), Panama City, Republic of Panama, **2** Universidad Americana de Europa (UNADE), Quintana Roo, México, **3** Universidad Internacional de la Rioja, La Rioja, Spain

‡ DQ and RGH are joint senior authors on this work.
* damian.quijano@udelas.ac.pa

## Abstract

We disclose a methodology to determine the participants in discussions and their contributions in social networks with a local relationship (e.g., nationality), providing certain levels of trust and efficiency in the process. The dynamic is a challenge that has demanded studies and some approximations to recent solutions. The study addressed the problem of identifying the nationality of users in the Twitter social network before an opinion request (of a political nature and social participation). The employed methodology classifies, via machine learning, the Twitter users' nationality to carry out opinion studies in three Central American countries. The Random Forests algorithm is used to generate classification models with small training samples, using exclusively numerical characteristics based on the number of times that different interactions among users occur. When averaging the proportions achieved by inferences of the ratio of nationals of each country, in the initial data, an average of 77.40% was calculated, compared to 91.60% averaged after applying the automatic classification model, an average increase of 14.20%. In conclusion, it can be seen that the suggested set of method provides a reasonable approach and efficiency in the face of opinion problems.

## 1. Introduction

Twitter has fewer users than other social networks, such as Facebook. Still, it could be said that Twitter has a disproportionate influence in the world, partly because it attracts a significant number of characters and individuals in the communication field, such as politicians, journalists, celebrities, public agencies, companies, and international organizations. Although there is a debate about the degree of confidence in predicting outcomes from the opinion of Twitter users, it is evident and manifest that the views and news on Twitter impact society. For example, social activism has grown through the hashtag. Among many others, the hashtag #BlackLiveMatter. [1], was used twelve million times and originated from the protests over the murder by a Minneapolis policeman of George Floyd, a man of African descent. The hashtag

**Data Availability Statement:** There are 30 tables stored in two repositories: repository 1 and repository 2. Repository 1 has 12 tables: https://github.com/damian-quijano/identify_nationality_twitter http://dx.doi.org/10.5281/zenodo.7254532

Repository 2 has 18 tables: https://github.com/
damian-quijano/-identify_nationality_twitter2 http://
dx.doi.org/10.5281/zenodo.7254534.

**Funding:** The authors did not received any funding
for this study.

**Competing interests:** The authors have declared
that no competing interests exist.

transformed dispersed initiatives into an organized movement through the social network. The distribution of information worldwide is no longer an exclusive service of international press agencies. For example, the first news about the death of Osama Bin Laden in Pakistan was published on Twitter by a local consultant [2]. When Hillary Clinton was a candidate for the Democratic Party, she announced that she would run for election via Twitter, leaving aside press releases or television appearances. Thus, Twitter was an important factor in changing the political dynamic. It justifies the enormous interest of information professionals, opinion researchers, sociologists, social network analysts, and data scientists in studying social networks.

On the other hand, the concept of viral news shows a new reality regarding the dissemination of information. Its most striking feature is that viral news can arise from any Twitter user [3], not necessarily from well-known individuals or institutions. Another important example of the influence of social networks is the impact on society of misinforming or fake news [4].

Therefore, although it cannot be stated with total certainty that the opinions of Twitter users influence the results of an election or decisions of national reach, it is undeniable that they generate debate and create a perception in society. Usually, government agencies, international organizations, politicians, and companies have chosen to publish their messages on this social network to announce and inform the community and analyze behaviors based on these communicational interactions.

The first challenge for national opinion studies is to gather a sample of users exclusively from the country trait. Twitter does not provide information about the user's country automatically. It is a user's choice. As stated in Twitter's privacy policies [5], the IP addresses of each message are stored. This information is not public. The user can display the following information publicly: the geographic coordinates when posting a tweet or adding information about their country in the user profile, specifically in the location and description sections. Still, most users of the network ignore it, resulting in a challenge for scholars of the social network Twitter. In general, this problem is solved by checking the nationality of each user, either by analyzing each of their messages, reviewing the profile description, observing the followers, following or interviewing the user. Thus, it consumes a lot of work. In this study, the concept of country trait is differentiated from geographic location. The geographic location is recorded in the tweet when the user marks her/is location when s/he posted the tweet. This does not necessarily mean that the user is a citizen of the country to which the geographic coordinates of the message belong. The user may be a tourist or a resident foreigner. Possibly the repetition of geographic coordinates of a large portion of tweets points to the country trait of a user, but it requires a large number of geotagged messages, which does not occur frequently.

On the other hand, the country trait does not necessarily depend on the geographical location of the tweets. A user of a country may be in transit or resides in another country, but it does not prevent him/er from being able to be part of the opinion debate of her/is own country on Twitter. S/he is a user who can even exercise the vote in his country's elections from other countries. Therefore, for researchers, the opinion of users of a country residing in another country is also essential. The concept of country trait has been defined as the attribute that identifies the country in which the user shows attachment and interest in participating and interacting with other users from the same country, regardless of their geographic location.

This study covers three Central American countries: Panama, Nicaragua, and Costa Rica. The results of the survey show that from the sampling and inference of the downloaded data from Twitter, among the three countries, an average of 22.66% of geotagged tweets do not reflect the users' nationality or they would be of a different nationality. Thus, the data maintains a high degree of mixture between those users identified and those who are not or are from other countries.

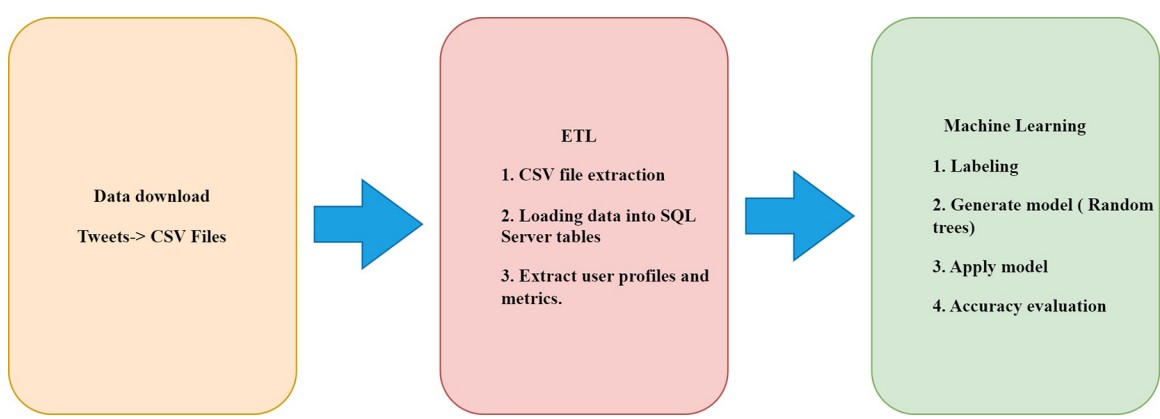

**Fig 1. Methodology stages summary.**

This project proposes a methodological approach to achieve a sample of country-trait users with a low percentage of mixing using three chained methods, as shown in Fig 1.

It is essential to know the structure of tweets and user profiles when the study was conducted (Twitter usually makes modifications periodically) to understand the differences between the research method and other related studies. Twitter provides information through tweets and user profiles. Two different packages of data are downloaded with the Twitter API [6]. Tweets are bodies of text that are internally structured in different sections. In addition to the text (the most visible part of the social network) of the tweet, more information can be extracted through the Twitter API. Below (Fig 2) are some of these elements): that are part of a

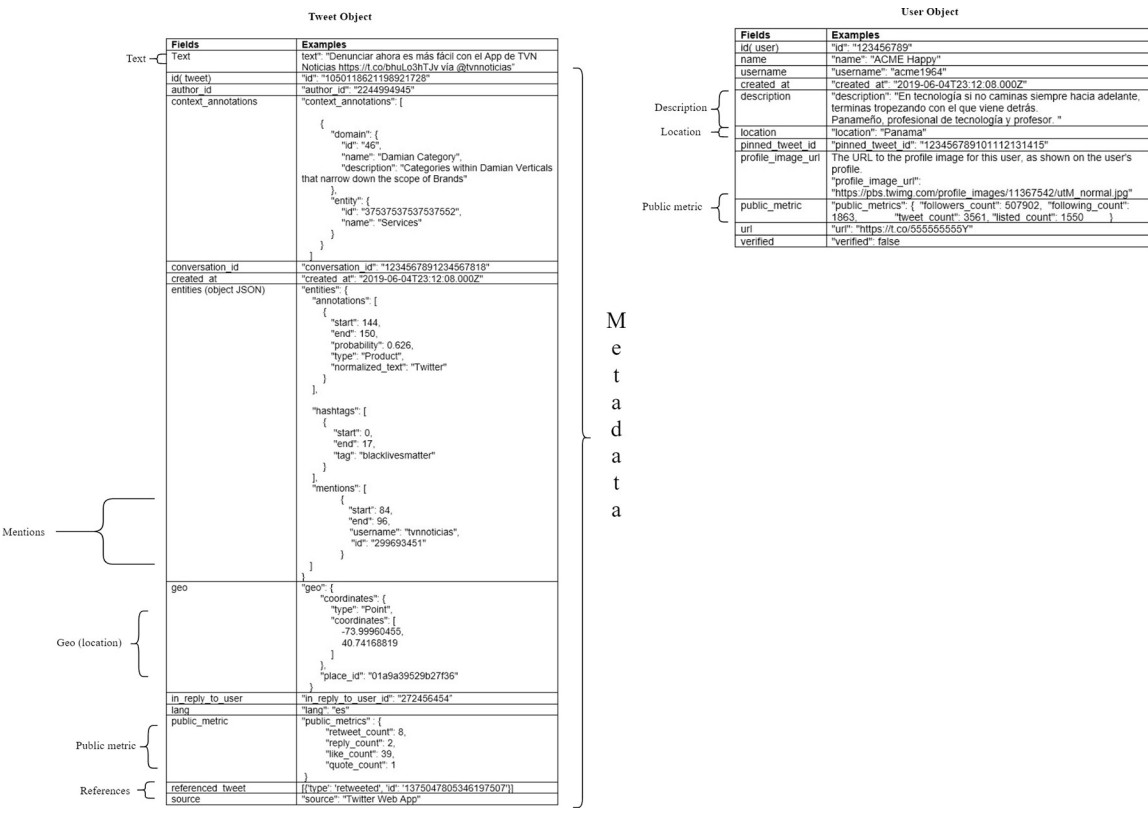

**Fig 2. Parts of the tweet and user object data structures.**

tweet *(belong to Tweet object)* and will be called fields: id (identifier) of the tweet, the id of the user, date of creation of the tweet, entities that are part of the text (characters, places, products, and organizations), topics on which the text is focused (for example animals, politics or sports), references to other sources (hashtags and URLs), references to other users (replies, mentions, retweets, and quotes), geographic location at the time of publishing the tweet, the language of the text, source (android, iPad, etc.) and public metrics that are the number of times that other users retweet, likes, replies, and quotes to the tweet, all this information other than the text of the tweet, is called metadata, Twitter calls this structure Tweet Object [7]. On the other hand, Twitter keeps a *record of information about each user*. This information contains the profile that is composed of the following fields: id (identifier) of the user, username, name, description or biography, location, website, a tweet posted, verified, the URL of the profile image, and public metrics (number of followers, number of followers, number of tweets published and number of lists), this structure is called *User Object* by Twitter [8], and allows for a deeper understanding of each user's data.

In Fig 2, shows some sections of information from both structures, indicating the fields of greatest use for social network studies. In the Tweet Object (left side), the following elements are shown: text, mentions, geographic location, public metrics, and references.

Mentions refer to the users that are mentioned in the tweet. References are of three types: replies, retweets, and quotes. The reply indicates that the tweet is a reply to another user. The retweet indicates that the tweet is the divulgation of another tweet, and the quote is the same as the retweet, but a comment is included. In these fields, the id of the referenced tweet and user is added.

The public metrics of the tweet object show the number of times it was replied to, retweeted, quoted, or liked by other users. For the case of the user object, studies usually analyze the description, location, and public metrics fields. The user sometimes adds clues about his country's traits in the description. In the location field, the user is expected to add his country, but as mentioned above, it happens rarely. The public metrics in the user object show the number of followers, follows, tweets, and user lists.

Different options of the Twitter() API are used to download the information of the tweet and user objects, as well as the lists of followers, followers, and likes.

The user location inference studies are based on three sources of information from the tweets and user accounts [9]: the content of the message (text of the tweet), the network of users (the metadata referring to followers and followed), and the context referring to the rest of the metadata [9, 10]. The message content analysis is usually performed by natural language processing (NLP) techniques [11] or classification models based on tokenized text corpora (text mining). User network analysis is based on relationships between users and it is usually treated with algorithms based on graph theory [12]. Finally, the context-based analysis identifies entities mentioned in the text, coordinates (if messages are geotagged) or user metadata information that allows inference attempts to be made; it is a hybrid approach that also apply machine learning techniques [13].

## 1.1 Related works

In the following, some studies that propose different strategies and analyses for location inference are mentioned to show the difference with the method suggested in the research.

In the study of [14] demographic information was obtained from the user's Twitter profile. In the profile, different information elements are displayed: description, user name, banner, pinned tweet, and the user can directly add her/is nationality. Sometimes users add references to their nationality in the report, e.g., country name, locality, province, or municipality.

Another study [15] proposed a probabilistic framework for estimating the city-level location of a Twitter user based solely on tweet content. The system calculates k possible locations for each user in descending order of confidence. These cited authors found that the location estimates converge quickly (needing only 100s of tweets), placing 51% of Twitter users within 100 miles of their actual location. Likewise, [16] used machine learning methods to deduce user demographic information from raw messages posted on Twitter. In its approach, it uses the Google Maps Geocoder API to map GPS coordinates, as well as to query the country in which a town/city/district resides. In [17], estimate a user's location at the city level based solely on the content of tweets (A Content-Based Approach), which may include tweet reply information. They compute a baseline probability estimate of the distribution of words used by a user. Pontes et. al. [18] analyze whether a simple method can infer the user's home location using publicly available attributes and geographic information associated with reachable friends. They found that it is possible to guess the user's home city with high accuracy, about 67%, 72%, and 82% of the cases in Foursquare, Google+, and Twitter, respectively. Gritta et al. [19] focused on geoparsing, which aims to translate free-text toponyms into geographic coordinates.

In each cited study, the concepts: of location, geolocation, coordinates, and localization are repeated. However, none of them guarantees that a tweet geolocated in a country implies that the user is from that country. This happens when the people who own the account are in transit through a country or are resident foreigners with no interest in the internal politics of the country of their residence. For example, in the case of Panama, there may be an important Panamanian community outside the country that has an opinion and influence on the results of a vote at the national level. On the other hand, there may be people who make their message heard in Panama but happen to be foreigners or residents who are not immersed in Panamanian culture. For a political opinion researcher, who wishes to predict results or measure perception, it is also of interest to identify nationality, even above location. Other studies attempt to infer nationality through a user-network approach such as that proposed by [20] from the mentioned network of popular users in a country. They used the tag propagation algorithm on the mentioned network to infer the locations of other users with unknown locations. The method was combined with the text-based geolocation inference method. In contrast, [21] opted for using social networks based on the following relationships, using different tag propagation algorithms based on graph theory to infer those untagged accounts.

Studies similar to our research focus on geographic location rather than identifying the country trait; possibly, there is a perception that both are strongly correlated. However, at least for the three countries analyzed in the study, we could verify that the percentage of nationals in the sample does not exceed 78% average, even though users tagged tweets in the country that is supposed to be of their origin or residence. The location is a geographic attribute, but the country trait is an attribute that represents a social condition, so the research has shown a particular interest in those studies that focus on automatically classifying characteristics of social condition, for example: knowing demographic attributes (age, employment status, social class), identify categories of tweets (for example fake news), types of accounts (bots, government, professional guild, and so on), or the existence of communities, all these studies are similar to the effort to identify the country trait of users.

The study focused on identifying users who show traits of identification with the country they consider of their genuine attention or of greater interest, under the assumption that by offering such traits, they can be classified as part of the culture, experience, and dialogue of that country in the social network Twitter. The first download of geotagged tweets allowed the grouping of users who, in principle, are of the nationality indicated by the geotagged information in the messages. However, the study showed that approximately 22.67% of the users are

not from the country or it is not possible to identify their nationality from the content and metadata of the messages and the user's profile.

The novelty of the research is to propose a method of automatic identification of the nationality of Twitter users through a classification model whose characteristics contain the number of times that different interactions occur between the users who are authors of the downloaded tweets and other metrics included in the metadata of the messages and user profiles, this approach is different from the current proposals based on inference from the textual contents of messages and profiles or through network analysis [9].

The study uses the machine learning technique called Random Trees to generate the national identity classification models. Random forests are composed of groups of random trees [22]. That works well for highly nonlinear relationships. The data distribution does not need to be parametric, normalization or standardization of values does not need to be applied, models are not affected by outliers, over-fitting of results is avoided, and the importance of the features used during model training can be known [23]. It allows the use of multiprocessing by processing in parallel each variant and estimator of the forest and is highly configurable.

In [24] and this proposal demonstrates that applicable classification models can be generated from a random sample of no more than 385 records. These criteria were tested during the development of the study with good results.

Much literature on research uses random trees to classify attributes or traits of interest. In detecting diabetes [25], uses the symptoms that the patient may manifest to identify the disease using machine learning techniques. They tested different algorithms: logistic regression, support vector machines: linear and nonlinear kernel, random forest, decision tree, adaptive boosting classifier, K-nearest neighbor, and naive Bayes. Random Forest achieved the best results with 98% accuracy, using 80% of the data set for training and 20% for testing. The study of [26] proposes a Random Forest classifier that automatically determines which tweets are from Syrian refugees and produced a promising 81% correct classification result. On another topic [27], addresses the problem of the effectiveness of anti-bird devices intended to reduce transmission line failures caused by bird activity. They applied a model generated from the Random Forests algorithm to predict the efficacy of an anti-bird device before its installation. On the well-known credit card fraud detection problem [28], uses two algorithms: the decision tree and the random forest. The first one identifies the user's activities and classifies those that are suspicious. In the second method, the suspect is identified and classified.

The study's method substantially reduces the effort of initial labeling for training data. It does not require costly processing resources since it uses a random sample of no more than 385 previously labeled records. 80% of the samples were for training and 20% for testing, allowing independent researchers or groups of researchers with limited resources to achieve a sample of sufficient size to start their opinion studies on the Twitter social network. The model that automatically classifies the rest of the users is generated from this sample.

The contributions of this study are:

- The validity of using small samples during the manual labeling process to construct training and test data is demonstrated.

- The results show that, from the sampling and inference of the downloaded data, about 22.66% of the geotagged tweets do not reflect the user's nationality.

- Retweets and quotes were the most influential features in the classification model in the three countries studied.

- Exclusively numerical features are used in the automatic classification model.

- The fact that the model's features do not depend on textual content makes it generalizable to any other language, the only difficulty being accurately identifying the nationality of the users when labeling a small sample.

- The Methodology does not depend on the number of tweets but users; tweets are only used to extract users.

This manuscript is organized as follows. Section 2 describes the Methodology, covering the data sets, the labeling process, and the model setup. The following section, Section 3, presents the results and discusses them, while Section 5 summarizes the paper in the Conclusion.

## 2. Methodology

Fig 3 shows the Methodology used in the study. The figure shows the steps of the methodology, the associated data tables and the results for the case of Panama. The same steps were applied for each of the three study countries.

A first download of tweets was made that were geotagged, then the users were extracted. A second download of tweets from previously saved users was made, but without limiting them to be geotagged, users were again extracted from the tweets. Mentions, replies, retweets and quotes were also separated to calculate the times they occur for each user, this allows adding new calculated fields to the users table. Then a random sample was extracted from the user

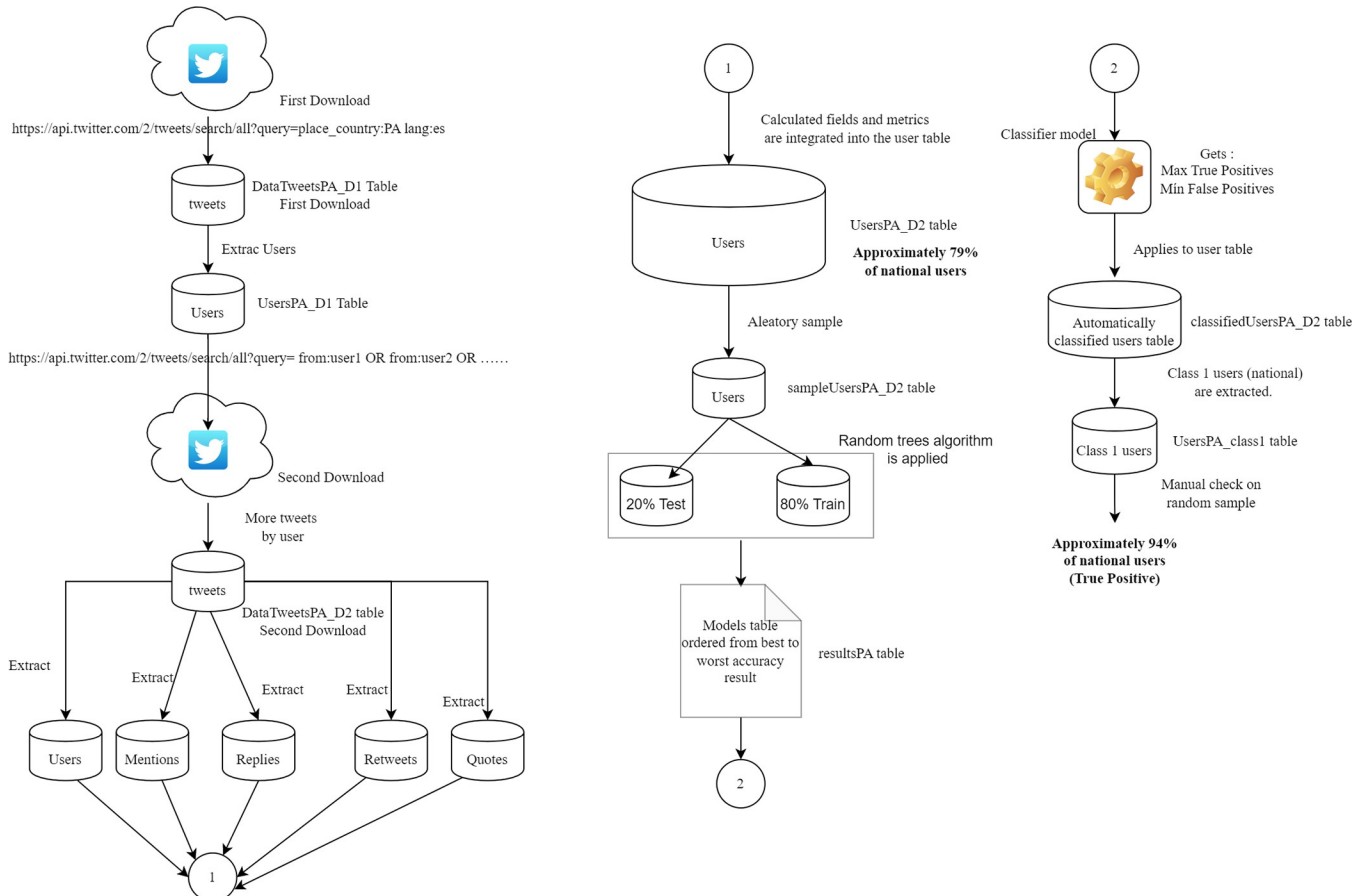

**Fig 3. Methodology data flow diagram used in the study, Panamá case.**

table and the random tree algorithm was applied with different combinations of characteristics and parameters, the different models were saved in a results table. The random tree algorithm was applied, a model was chosen that achieved the lowest false positive and the highest true positive. This model was applied to the user table to predict nationality (1 is Panama, 0 is not Panama). Finally, a random sample was taken from the table of automatically classified users and compared with the real values verified in the social network.

## 2.1 Software

The study was delimited to Central American countries because of cultural affinity, language, and history. The researcher version of Twitter's application programming interface (API) was used to search and download tweets and data for each account [29]. The database used to contain the information and data analysis was Microsoft's SQL Server [30]. Python programming language was used [31], version 3.9, from the Spyder 5.0.5 IDE environment, installed via Anaconda (https://www.anaconda.com/products/individual). Work was performed on a laptop computer Intel(R) Core(TM) i7-8750H CPU @ 2.20GHz 2.21 GHz,16.0 GB (15.7 GB usable), 64-bit operating system, x64-based processor, OS. Windows 10, 1GB asynchronous internet access, a 500GB solid-state drive (SSD), and another 1TB HDD.

## 2.2 Data collection

The study data mainly used 30 tables stored in two repositories:

Repository 1 has 12 tables: https://github.com/damian-quijano/identify_nationality_twitter
http://dx.doi.org/10.5281/zenodo.7254532

Repository 2 has 18 tables: https://github.com/damian-quijano/-identify_nationality_twitter2
http://dx.doi.org/10.5281/zenodo.7254534

Large tables have been saved in the first repository, small size tables and code have been saved in the second repository. All the tables mentioned in the article can be consulted in the repositories.

In the first download of tweets the place_country criterion [32] was included in the query. Using the search/all API [33], tweets were downloaded under the following criteria (Panama PA case):

*https://api.twitter.com/2/tweets/search/all?query=place_country:PAlang:en&start_time= 2021-01-01T00:00:00:00.000Z&end_time=2021-12-31T00:00:00:00.000Z&max_results= 500&tweet.fields=created_at,source,conversation_id,in_reply_to_user_id,public_metrics, referenced_tweets,entities&user.fields=verified&expansions=author_id*.

For the case of the rest of the countries, the value of place_country is replaced. The limits set by Twitter for tweet downloads must be taken into account. In the case of the API search/ all Researchers version, it allows the launching of a maximum of 300 requests in no more than 15 minutes. Each request downloads a maximum of 500 tweets. From the first request for tweets, a 15-minute window is activated. If the 300 requests are sent within 15 minutes, you have to wait (pause the program) for the end of that time to launch a new round of 300 new requests.

Each request downloads 500 tweets, which are saved in a delimited "CSV" file. Once a total load of messages from the country was completed, all the.csv files were integrated into one file, then the complete file was added to the database as a Table that contained 4,400,334 de tweets. The results for each country are shown in Table 1.

**Table 1. Number of tweets downloaded (first time) by each central American Country.**

| Country | Tweets |
|---|---|
| Panamá(PA) | 1,236,261 |
| Guatemala(GT) | 917,048 |
| Costa Rica(CR) | 727,169 |
| El Salvador(SV) | 591,711 |
| Honduras(HN) | 547,655 |
| Nicaragua(NI) | 380,458 |

To know which are the most used devices and software at the time of geotagging tweets, the number of messages was grouped by the source. This refers to the devices or software recorded in the tweet information.

In Fig 4 more than 94% of the geotagged messages were published from cell phones or smartphone.

From the messages downloaded the first time, users were extracted, with the results shown in Table 2.

The user data was stored in the tables: UsersPA_D1, UsersGT_D1, UsersSV_D1, UsersCR_D1, UsersHN_D1 and UsersNI_D1, which will be mentioned as UsersCountry_D1. Users who are the authors of the posts or actions (replicate or mention) of the downloaded tweets refer to them as usersIN. The user data does not include users who are replicated or mentioned. The users mentioned or referred to in the tweets but are not part of the users who are the authors of the downloaded messages were called UserOut, they have a different treatment that will be explained later.

As will be seen below, the number of tweets downloaded with the place_country option is minimal compared to the total number of messages that users post.

Given the monthly download limit applied by Twitter (10 million tweets), it was necessary to adjust the download time interval, which started on January 1, 2021, until March 31, 2021. It was finally decided to select three countries: Panama (most significant number of users), Costa Rica (average number), and Nicaragua with the smallest number of users. These limits allowed not to exceed more than 10 million tweets downloaded per month.

| Panamá | | Costa Rica | | Guatemala | | Honduras | | Nicaragua | | Salvador | |
|---|---|---|---|---|---|---|---|---|---|---|---|
| source | % | source | % | source | % | source | % | source | % | source | % |
| Twitter for Android | 58.81% | Twitter for Android | 54.56% | Twitter for Android | 69.83% | Twitter for Android | 58.64% | Twitter for Android | 67.45% | Twitter for Android | 76.49% |
| Twitter for iPhone | 35.52% | Twitter for iPhone | 41.22% | Twitter for iPhone | 28.00% | Twitter for iPhone | 39.36% | Twitter for iPhone | 29.93% | Twitter for iPhone | 20.77% |
| Instagram | 5.48% | Instagram | 3.02% | Instagram | 1.93% | Instagram | 1.94% | Instagram | 2.49% | Instagram | 2.49% |
| Others | 0.19% | Others | 1.21% | Others | 0.24% | Others | 0.07% | Others | 0.13% | Others | 0.25% |

**Fig 4. Usability percentage by type used devices.**

**Table 2. Number of users for each central American country.**

| Country | Users |
|---|---|
| Panamá(PA) | 17,292 |
| Guatemala(GT) | 14,314 |
| El Salvador(SV) | 13,642 |
| Costa Rica(CR) | 12,000 |
| Honduras(HN) | 8,991 |
| Nicaragua(NI) | 6,942 |

| Panamá | | Costa Rica | | Nicaragua | |
|---|---|---|---|---|---|
| source | % | source | % | source | % |
| Twitter for Android | 52.87% | Twitter for Android | 45.84% | Twitter for Android | 61.05% |
| Twitter for iPhone | 34.89% | Twitter for iPhone | 36.84% | Twitter for iPhone | 23.99% |
| Twitter Web App | 6.87% | Twitter Web App | 8.02% | Twitter Web App | 6.75% |
| Instagram | 1.67% | Instagram | 2.36% | objectpost | 1.65% |
| TweetDeck | 0.83% | Twitter for iPad | 1.08% | Instagram | 1.35% |
| Twitter for iPad | 0.70% | SOTAwatch | 0.93% | TweetDeck | 1.07% |
| EarthquakeTrack.com | 0.26% | objectpost | 0.82% | EarthquakeTrack.com | 0.94% |
| Buffer | 0.20% | TweetDeck | 0.61% | CarpoolWorld Feed | 0.55% |
| twitbot_h | 0.18% | EarthquakeTrack.com | 0.47% | EMSC Felt earthquakes | 0.51% |
| Others | 1.54% | Others | 3.03% | Others | 2.15% |

**Fig 5. Comparison of devices used in the three countries analyzed in the study.**

Having the names of the user accounts, we started the download of the messages of each user again without including the search criterion place_country to download all the messages of the users using the following command request:

*https://api.twitter.com/2/tweets/search/all?query=from:user1%20OR%20from:user2%20OR........&start_time=2021-01-01T00:00:00.000Z&end_time=2021-03-30T00:00:00.000Z&max_results=500&tweet.fields=created_at,source,conversation_id,in_reply_to_user_id,public_metrics,referenced_tweets,entities&user.fields=verified&expansions=author_id*

Up to 40 users are included in each request. This made it possible to download all messages for each user, not only those posted from their smartphone and geotagged.

It was found that the number of messages is much higher than those downloaded the first time.

The results of the downloads (second time) for the three countries are as follows:

- Panamá: 4,657,755 tweets. DataTweetsPA_D2 table.

- Costa Rica: 2,551,889 tweets. DataTweetsCR_D2 table.

- Nicaragua: 1,272,375 tweets. DataTweetsNI_D2 table.

In order to find out again which are the most used equipment and software when publishing the tweets, a grouping of the number of messages by source was made. In Fig 5 shows more than 81% of the messages were posted from smartphones.

### 2.3 ETL process

From the downloaded messages (second time), the users are extracted again, as follow:

Panamá (PA) 14,789 users UsersPA_D2 table

Costa Rica(CR) 9,843 users UsersCR_D2 table

Nicaragua(NI) 5,223 users UsersNI_D2 table

The data saved in the CSV files were extracted and loaded into UsersPA_D2 table, UsersPA_CR tabla, and UsersNI_D2 table; UsersCountry_D2 will be mentioned when referring to all of them. There is a decrease in the number of users on each table; this has different explanations: the number of months in the search for tweets was reduced; therefore, there will be new users that did not yet exist in the months selected for the study (January, February, and March 2021). On the other hand, there are frequent interventions in the status of accounts and messages by the Twitter administration, specifically regarding blocking, deletion, and restriction actions to accounts and messages, which implies that there may be a variation in the measurements made in different times.

From the downloaded tweets, were extracted separately messages, mentions, and references [7] for each country and stored on separate tables. The table of references classified their different types: retweets, replies, and quotes.

Using the User Lookup API [34], the following user data was downloaded from the information structure contained in each user account [8]:

- created_at: date of account creation. Categorical type.

- Verified: account verified or not. Categorical type.

- Followers: number of followers. Integer type.

- Following: number of followers. Integer type.

- tweet_count: number of tweets since account creation. Integer type.

- listed_count: number of lists created by the account. Integer type.

Columns representing calculated metrics/indicators from the number of messages and interaction actions between users have also been added to each user in UsersCountry_D2 tables:

- cant_tweets_show: is the sum of all the user's tweets contained in the set of downloaded tweets. It is not equal to the tweet_count field, which is part of the user's profile (along with followers, following, etc.) and shows the number of tweets posted since the account was created. Integer type.

- Rt: is the sum of all retweets applied to all the user's messages, messages contained in the set of downloaded tweets. Integer type.

- Retweets: is the sum of all the retweets applied to all the user's messages and messages contained in the set of downloaded tweets—integer type.

- Likes: the sum of all likes applied to all the user's messages and messages contained in the set of downloaded tweets—integer type.

- rquotes: is the sum of all retweets with quotes that were applied to all the user's messages and messages contained in the set of downloaded tweets. Integer type.

The mentioned or referenced users can be either usersIN, owners of the downloaded messages, or UsersOUT, they are not owners of the downloaded messages. In the case of quote-type references, the tweets do not provide the information of the user making the quote, only the referenced tweet id (privacy purpose). This implied downloading all the referenced tweets to identify the author account of the publication/post and download its data and identify the UsersOut. New fields were added to the UsersCountry_D2 tables containing the amounts of the following actions among users: number of tweets, mentions, replies, retweets, and quotes. The structure of each UsersCountry_D2 table is presented as follow:

- author_id. Large integer type.

- Username. Literal type.

- created_at: date of account creation. Type date.

- Verified: account verified or not. Type literal.

- Followers: number of followers. Integer type.

- Following: number of followers. Integer type.

- tweet_count: number of tweets since account creation. Integer type.

- listed_count: number of lists created by the account. Integer type.

- cant_tweets_sample: number of tweets that are part of the downloaded sample. Integer type.

- Rt: number of retweets to tweets from the account that are part of the downloaded sample. Integer type.

- vreplies: number of retweets to tweets from the account from the downloaded sample. Integer type.

- Likes: number of likes to tweets of the account from the downloaded sample. Integer type.

- rquotes: number of retweets commented (quotes) to tweets of the account that are from the downloaded sample—integer type.

- mentions_a_In: number of mentions made by the user to other users that are part of the downloaded sample. Integer type.

- mentions_a_Out: number of mentions made by the user to other users that are not part of the downloaded sample. Integer type.

- mentions_of_In: number of mentions made by users (that are part of the downloaded sample) to the user of the message. Integer type.

- rt_a_In: number of retweets made by the user to other users that are part of the downloaded sample. Integer type.

- rt_a_Out: number of retweets made by the user to other users not part of the downloaded sample. Type integer.

- rt_de_In: number of retweets made by users (that are part of the downloaded sample) to the message user. Type integer.

- rp_a_In: number of retweets made by the user to other users that are part of the downloaded sample. Integer type.

- rp_a_Out: number of replicas made by the user to other users not part of the downloaded sample. Type integer.

- rp_de_In: number of replicas made by users (who are part of the downloaded sample) to the user of the message.

- rq_a_In: number of quotes made by the user to other users who are part of the downloaded sample. Type integer.

- rq_a_Out: number of quotes made by the user to other users who are not part of the downloaded sample. Type integer.

- rq_de_In: number of quotes made by the user to other users that are not part of the downloaded sample. Type integer.

- Activity: sum of actions (rt, RP, RQ, and mentions) performed by the user. Integer type.

- Localization: user's location information is displayed in his profile if the user decides to write it. Literal type.

- Description: description of the user displayed in his profile, in case the user decided to write it. Literal type.

- description_depurada: description of the user shown in their profile, in case the user decided to write it, but graphic symbols were removed. Literal type.

- profile_link: user's a link on Twitter. Literal type.

## 2.4 Measuring the proportion of national users

Sampling was performed to calculate the proportion of accounts that can be identified as national. The sample size was calculated assuming the values p = q = 0.50, the maximum possible, for the confidence of 95% and sampling error of 5%, the size is 385 users.

The program to create the random sample (Repository 2, http://dx.doi.org/10.5281/zenodo.7254534) of the total quantity of messages for each country was written in Python by Quijano author, using the Pandas library [35].

### 2.4.1 Manual tagging

Different situations types were observed when identifying the nationality of users during the labeling process:

1. Users who geotagged in their profile clearly the Country or locality of the Country of interest.

2. Users who geotagged in their profile clearly a Country or locality, but it is not the country of interest.

3. Users who did not geotag, but the country of interest is clearly and quickly identified by other information, for example, by including an allusion to their country in the profile description or showing their flag image.

4. Users who were not geotagged, but the country is clearly and quickly identified by other information, even though it is not the country of interest.

5. Users who did not geotag and have not been able to identify a Country (at least not clearly and quickly).

Type #1 and type #3 were labeled with Y (the country of interest is identified), and the rest are N (not the country of interest or not identifiable).

Two categories were established: accounts whose nationality is identifiable (category S) and those whose nationality is not of the corresponding country or is not identifiable with full certainty (category N).

Listed below are elements usually included in the tweet and profile of accounts that allow their nationality to be identified.

- Geo tagging in the profile. The name of the country or national locality is displayed.

- Flag in the profile text.

- Flag, a reference to the country or a known Country location or province, phone, site, or person from the country is shown in the description, banner, or pinned tweet of the profile.

- Country pre or suffix included in the username.

- In addition to the Country name, provinces and localities or municipalities or addresses are also often displayed.

Most of the accounts do not include the information listed above. In those cases, we chose to read the user's first 10 most recent tweets and study the following elements:

- Expressions that allow the identification of nationals, for which a list of such expressions were consulted.

- Mentions of political figures or actions announced or carried out by governments and their agencies.

- Sports topics, specifically soccer, in which reference is made to the local or national soccer club.

- Complaints with Internet providers that can be clearly identified as being part of the country.

- Political references.

- Mentions of events of a large attendance, for instance, the presentation of the musical band Coldplay in Costa Rica.

Those users whose nationality could be clearly and quickly identified were labeled with the value 1. Otherwise, they were marked 0. The users classified with category 0 showed two different results: one group of users whose country could be clearly and quickly identified, but was not the country of the study, and the other group whose country could not be clearly and quickly identified. This last case is frequent in the following accounts: accounts dedicated to sexual services, accounts dedicated to promoting celebrities, fan accounts, organizations with international attention or reach very little active or recent accounts, and generally very young or young users.

Next, the users of the sample were manually classified. The labeling was performed by three analysts who carried out the following steps. In the beginning, a sample selection of 385 users for a Country was made and given to the three analysts to start a test classification. In the end, the three compared their results and agreed on the differences. The purpose of this first classification is to familiarize the analysts with the process of early and clear identification of nationality or to establish that it is unidentifiable criteria. Next, another sample of 385 users was given to each of the three to label the nationality if possible. They then compared the results and tried to agree on the differences; otherwise, they decided by voting—the same for the samples of the rest of the countries.

Table 3 shows that users whose nationalities were identified in the samples do not exceed 79% in any of the countries.

**Table 3. Inference of the statistic proportion of users whose nationality was clearly and quickly identified.**

| Random sample size | Panamá | Costa Rica | Nicaragua |
|---|---|---|---|
| 380 | 79.34% | 76.56% | 77.15% |

## 2.5 Construction of classifier models using random trees

Since the study intends to extract as many users of the nationalities of interest as possible from the tweet download automatically and to reduce the number of users not identified or of a different nationality than the country under study, we chose to use random trees to generate a classifier model that does not depend on normal or parametric distributions and that works well with small training samples.

The python library sci-kit-learn [36] was used to process the data using the automatic learning technique Random Forests, which from now on we will call RF, is a supervised technique that, from a set of labeled data, generates a classifier model to predict the target value of the unlabeled data [24].

The study, from a sample of 385 users, generates a classifier model that allows classifying the rest of each country's data to increase the precision in identifying national users. For a better explanation of the method, the results for Panama will be shown. UsersPA_D2 table has 14,789 users, and according to the sampling results, it is inferred that 79.34% of users are Panamanian. The rest of the users are part of other countries, or their national identity is not identifiable. The aim is to increase the percentage of accuracy of identification of Panamanian accounts, at least up to 90%.

The classifier models have different accuracy metrics, so the confusion matrix is used [37] for such a measurement, it shows information that allows the information of interest to be identified. The confusion matrix shows whether the accuracy reflects reality by comparing the prediction results against the actual results. The confusion matrix presents four results: true negatives, false negatives, false positives, and true positives. In the study, two classes are the object of the classification: clearly identified national users and classified as class 1 or positive. Users of different nationalities or who could not be identified (because of their nationality) are classified as class 0 or negative. Therefore, according to the confusion matrix, it shows the following results:

- True negative (TN): when the model is correct that the user is not a national. The prediction (classification) is 0 and it really is 0.

- False Negative (FN): when the model is not correct, and the user is not a national. The prediction (classification) is 0, but it is actually 1.

- False positive (FP): when the model is wrong in identifying a user with a nationality of interest. The prediction(classification) is 1, but it is actually 0.

- True positive (TP): when the model is correct in identifying a user with a nationality of interest. The prediction(classification) is 1, and it is actually 1.

The goal of the usual metrics [38] is to maximize true positives and negatives, but it can happen that the false positive is very high, which, when averaged, yields a high but misleading measurement. In this study, trying to increase true positives as much as possible and decrease false positives to almost cero records, so that almost all users classified as national (class 1 or positive) are national, even though this implies an increase in false negatives and decreases the overall accuracy. The objective is to reduce the FP as much as possible, even if the TP decreases, and regardless of the TN and FN results, in this way a very pure TP sample is achieved, even if it is smaller.

An example of results are shown below to understand the intuitiveness of the intended goal. The example shows two confusion matrix results.

1. TN:20, FN:5, FP:20, TP:50. High TP and TN, but also a high FP.

2. TN:20, FN:30, FP:1, TP:30. Lower TP, but at the same time, very low FP.

Although there is a very high FP in the first example, so is the FP; therefore, there are 70 positives (users identified as national), of which there are 20 that, in reality, are not. This greatly increases the uncertainty when using the sample in an opinion study that requires the participation of exclusively national users. On the other hand, in the second example, there are fewer TP, only 30 users, but only 1 FP case. Despite having a smaller number of identified nationals, there is, therefore, the peace of mind to think that it is an almost entirely correct sample. Therefore, for an opinion study, there is greater certainty for the case of those 30 users. We use the precision metric/ indicators formula that calculates the ratio TP / (TP + FP), where TP is the number of true positives and FP is the number of false positives [39]. If the result reaches 1, the total number of correctly predicted nationals is 100%. However, this implies many false negatives, users classified as unidentified and who are nationals. In the first example, calculating TP / (TP + FP) = 50 /70 = 71.42%. In the second example the result is TP / (TP + FP) = 30 /31 = 96.77%.

Like all other Machine Learning algorithms, random trees have hyperparameters that can be tuned to achieve a model that meets the objectives of the study; therefore, it is necessary to find the values of the hyperparameters that, together with the correct features, reach a maximum value of true positives and a minimum of false positives. The GridSerachCV Library [40] accepts groups of different values of the hyperparameters to combine them in each random forest model and measure the accuracy of the results of each of them. This library also helps to apply cross-validation [41] to estimate model results using partitions for training and others for validation. In this study, the estimators used in GridSeach are random forests containing parameters that must be configured [24]. It was decided to leave the default values for most of the parameters, except for the following:

- n_estimators: number of trees in the forest, 10 were chosen. After many tests, no better results could be observed with a larger number of trees, so 100, 1000, and 2000 were tried. The advantage of a few estimators is the processing time, which is much less and provides the same results.

- Criterion: refers to the measure of the quality of each division in the tree. Two criteria were added: the Gini index, which measures the impurity of each node, and the entropy, which measures the information gain.

- class_weight: is applied for the case of unbalanced data, and there are three options: 0 unbalanced, 1 balanced, and 2 balanced_subsample. The data reveal a higher amount of nationals. This may affect the results; therefore, the parameters included the three options.

- random_state: a value, 123, was set so that the results can be reproduced.

The training data were the 385 records previously labeled and randomly selected from the complete data of a Country. In turn, they were divided into 80% training part and the remaining 20% for testing purpose by means of the instruction train_test_split [42].

A very important factor was to select the columns or variables (model features) that the classifier model uses for training. Random forests use decision trees to randomly assign a portion of rows and a group of features from the training data. The use of all Table variables does not imply higher model accuracy. Some features are redundant, and many possible combinations can be tried until the results of the confusion matrix show a high TP and a low FP.

There are different ways to select the characteristics or variables of the model [43]. However, the best result was obtained by using "brute force", that is, training the data with many combinations until the best result was found. The Users2-Pais Table have 22 quantitative

variables. The following possible combinations of actions (mentions, references, and replicas) grouped in Python lists used for training were established. The list of possible combinations with the columns on mentions is shown below.

menciones = [

 ["","",""],

 ["","menciones_a_Out",""],

 ["","menciones_a_Out","menciones_de_In"],

 ["","","menciones_de_In"],

 ["menciones_a_In","",""],

 ["menciones_a_In","","menciones_de_In"],

 ["menciones_a_In","menciones_a_Out",""],

 ["menciones_a_In","menciones_a_Out","menciones_de_In"]

 ]

The list shows combinations in which none of the features that have to do with the mentions action ["","",""] are included, or only the mentions_to_Out action ["", "mentions_to_-Out",""], or all the mentions columns ["mentions_to_In", "mentions_to_Out", "mentions_of_In"] and so on until all possible combinations for the mentions action are covered.

Below are the list groupings for retweets, replies, and quotes.

retweets = [

 ["","",""],

 ["","rt_a_Out",""],

 ["","rt_a_Out","rt_de_In"],

 ["","","rt_de_In"],

 ["rt_a_In","",""],

 ["rt_a_In","","rt_de_In"],

 ["rt_a_In","rt_a_Out",""],

 ["rt_a_In","rt_a_Out","rt_de_In"]

 ]

replicas = [

 ["","",""],

 ["","rp_a_Out",""],

 ["","rp_a_Out","rp_de_In"],

 ["","","rp_de_In"],

 ["rp_a_In","",""],

 ["rp_a_In","","rp_de_In"],

```
    ["rp_a_In","rp_a_Out",""],

    ["rp_a_In","rp_a_Out","rp_de_In"]

  ]

rquotes = [

   ["","",""],

   ["","rq_a_Out",""],

   ["","rq_a_Out","rq_de_In"],

   ["","","rq_de_In"],

   ["rq_a_In","",""],

   ["rq_a_In","","rq_de_In"],

   ["rq_a_In","rq_a_Out",""],

   ["rq_a_In","rq_a_Out","rq_de_In"]

  ]
```

The possible combinations add up to 4,096 different options when using the columns that record user actions: mentions, retweets, replies, and rquotes. To this, we must add other actions that were grouped as follows:

["followers", "following", "tweet_count", "listed_count"], are metrics that are not influenced by the tweets downloaded. These metrics are a product of the cumulative numbers as of the date of the query and include all posts and sums for each user, not limited to downloaded posts. When opting to include these fields in the combination, all 4 columns are included. Otherwise none.

[rt", "vreplicas", "likes", "rtquotes"], are metrics that are not part of the interaction between users and authors of the tweets downloaded or calculated by the research, the values of these metrics are the product of the consultation of the information incorporated in the structure of each tweet. To this group was added ["cant_tweets_sample"]. When opting to include these fields in the combination, all 4 columns are included. Otherwise none.

[Activity"], this computed column adds up the number of shares totaled by the user. At the time of constructing the combinations, it can be included or not.

Therefore, there are a total of 196,420 possible combinations without repetition that are constructed with: the 22 columns of the UsersCountry_D2 tables, the two metrics of gini and entropy measurement, and the three balancing options.

Using a self-developed program (Repository 2, http://dx.doi.org/10.5281/zenodo.7254534), the random forest algorithm was applied to each of the combinations without repetition and being estimators of the gridSearchCV module. The precision of the results of each combination was measured with the confusion matrix. The result for each country was 196,420 rows stored in the tables:resultsPA,resultsCR and resultsNI, which will be mentioned as results-Country. Each row of the tables included the FN, FP, TP, and TN values along with the rest of the results.

The column structure of each table is as follows:

• Num: the user's number.

• TN: true negative (a non-national is predicted and is correct)

- FP: false positive (a national is predicted and is false)

- FN: false negative (a non-national is predicted and is false)

- TP: true positive (a national is predicted and is correct)

- cols: shows the columns or characteristics that were removed for training the model; therefore, the ones that do not appear are those columns used by the model.

- Params: the parameters that were configured from the random forest.

- Grid: the parameters set to gridSearchCV.

- Best: the best parameters proposed by gridSearchCV.

**2.5.1 Evaluation of sample labeling.** TP, FP, TN and FN, they are values of the confusion matrix calculated from the Test data (20% = 77 records) of the random sample of 385 records. So, we have that 80% is the train part (308 records) and 20% is the test part (77 records), the confusion matrix checks the number of records that match or not the actual values in the test part when applying the model generated from the train part. Therefore, TP + FP + TN + FN = 77 (total test records).

Using the following SQL SELECT statement, applied to resultsCountry tables, the results of the confusion matrix with high TP and low FP were searched:

SELECT [num],[TN],[FP],[FN],[TP],[cols],[best] FROM resultsPA where FP<2 order by TP desc

Fig 6 shows the models (each model is a record of the table) the results of the query for Panamá. The first rows show the records that meet FP (false positives) < = 2 and a maximum value of TP (true positives) sorted by TP in descending order. The first rows show models without false positives. The cols and best columns store the features and parameters used by the model that achieved the row's TP, FP,T N, and FN.

The cols column provides information that allows us to know the characteristics used by the model that achieved the results of the first row:

Figs 7 and 8 show the results for Costa Rica and Nicaragua. In the case of Nicaragua, no results were produced where FP = 0, but results were found for FP = 1.

Table 4 shows the features for each country that provided the best results according to the metrics FP = 0 or 1 and maximum TP. It can be seen that the models used between 2 up to 4

| | num | TN | FP | FN | TP | cols | best |
|---|---|---|---|---|---|---|---|
| 1 | 163575 | 18 | 0 | 30 | 29 | followers,following,tweet_count,listed_count,cant_tweets_muestra,rt,vreplicas,li... | {'class_weight': 'balanced_subsample', 'criterion': 'gini', 'n_estimators': 10} |
| 2 | 163576 | 18 | 0 | 30 | 29 | followers,following,tweet_count,listed_count,cant_tweets_muestra,rt,vreplicas,li... | {'class_weight': 'balanced_subsample', 'criterion': 'gini', 'n_estimators': 10} |
| 3 | 196342 | 18 | 0 | 30 | 29 | followers,following,tweet_count,listed_count,cant_tweets_muestra,rt,vreplicas,li... | {'class_weight': 'balanced', 'criterion': 'gini', 'n_estimators': 10} |
| 4 | 196343 | 18 | 0 | 30 | 29 | followers,following,tweet_count,listed_count,cant_tweets_muestra,rt,vreplicas,li... | {'class_weight': 'balanced', 'criterion': 'gini', 'n_estimators': 10} |
| 5 | 65366 | 18 | 0 | 30 | 29 | followers,following,tweet_count,listed_count,cant_tweets_muestra,rt,vreplicas,li... | {'class_weight': 'balanced_subsample', 'criterion': 'entropy', 'n_estimators': 10} |
| 6 | 98133 | 18 | 0 | 30 | 29 | followers,following,tweet_count,listed_count,cant_tweets_muestra,rt,vreplicas,li... | {'class_weight': 'balanced', 'criterion': 'entropy', 'n_estimators': 10} |
| 7 | 65365 | 18 | 0 | 31 | 28 | followers,following,tweet_count,listed_count,cant_tweets_muestra,rt,vreplicas,li... | {'class_weight': 'balanced_subsample', 'criterion': 'entropy', 'n_estimators': 10} |
| 8 | 98132 | 18 | 0 | 31 | 28 | followers,following,tweet_count,listed_count,cant_tweets_muestra,rt,vreplicas,li... | {'class_weight': 'balanced', 'criterion': 'entropy', 'n_estimators': 10} |
| 9 | 98137 | 18 | 0 | 32 | 27 | followers,following,tweet_count,listed_count,cant_tweets_muestra,rt,vreplicas,li... | {'class_weight': 'balanced', 'criterion': 'entropy', 'n_estimators': 10} |
| 10 | 65370 | 18 | 0 | 32 | 27 | followers,following,tweet_count,listed_count,cant_tweets_muestra,rt,vreplicas,li... | {'class_weight': 'balanced_subsample', 'criterion': 'entropy', 'n_estimators': 10} |
| 11 | 196347 | 18 | 0 | 32 | 27 | followers,following,tweet_count,listed_count,cant_tweets_muestra,rt,vreplicas,li... | {'class_weight': 'balanced', 'criterion': 'gini', 'n_estimators': 10} |
| 12 | 163580 | 18 | 0 | 32 | 27 | followers,following,tweet_count,listed_count,cant_tweets_muestra,rt,vreplicas,li... | {'class_weight': 'balanced_subsample', 'criterion': 'gini', 'n_estimators': 10} |
| 13 | 163581 | 18 | 0 | 33 | 26 | followers,following,tweet_count,listed_count,cant_tweets_muestra,rt,vreplicas,li... | {'class_weight': 'balanced_subsample', 'criterion': 'gini', 'n_estimators': 10} |
| 14 | 163644 | 18 | 0 | 33 | 26 | followers,following,tweet_count,listed_count,cant_tweets_muestra,rt,vreplicas,li... | {'class_weight': 'balanced_subsample', 'criterion': 'gini', 'n_estimators': 10} |
| 15 | 196348 | 18 | 0 | 33 | 26 | followers,following,tweet_count,listed_count,cant_tweets_muestra,rt,vreplicas,li... | {'class_weight': 'balanced', 'criterion': 'gini', 'n_estimators': 10} |
| 16 | 196411 | 18 | 0 | 33 | 26 | followers,following,tweet_count,listed_count,cant_tweets_muestra,rt,vreplicas,li... | {'class_weight': 'balanced', 'criterion': 'gini', 'n_estimators': 10} |
| 17 | 65429 | 18 | 0 | 33 | 26 | followers,following,tweet_count,listed_count,cant_tweets_muestra,rt,vreplicas,li... | {'class_weight': 'balanced_subsample', 'criterion': 'entropy', 'n_estimators': 10} |
| 18 | 65434 | 18 | 0 | 33 | 26 | followers,following,tweet_count,listed_count,cant_tweets_muestra,rt,vreplicas,li... | {'class_weight': 'balanced_subsample', 'criterion': 'entropy', 'n_estimators': 10} |
| 19 | 98138 | 18 | 0 | 33 | 26 | followers,following,tweet_count,listed_count,cant_tweets_muestra,rt,vreplicas,li... | {'class_weight': 'balanced', 'criterion': 'entropy', 'n_estimators': 10} |
| 20 | 98201 | 18 | 0 | 33 | 26 | followers,following,tweet_count,listed_count,cant_tweets_muestra,rt,vreplicas,li... | {'class_weight': 'balanced', 'criterion': 'entropy', 'n_estimators': 10} |
| 21 | 98106 | 18 | 0 | 34 | 25 | followers,following,tweet_count,listed_count,cant_tweets_muestra,rt,vreplicas,li... | {'class_weight': 'balanced', 'criterion': 'entropy', 'n_estimators': 10} |

**Fig 6. Example of the results of resultsPA table, Panamá case.**

| num | TN | FP | FN | TP | cols | best |
|---|---|---|---|---|---|---|
| 62499 | 17 | 0 | 28 | 32 | followers,following,tweet_count,listed_count,ca... | {'class_weight': 'balanced_subsample', 'criterion': 'entropy', 'n_estimators': 10} |
| 65019 | 17 | 0 | 28 | 32 | followers,following,tweet_count,listed_count,ca... | {'class_weight': 'balanced_subsample', 'criterion': 'entropy', 'n_estimators': 10} |
| 95266 | 17 | 0 | 28 | 32 | followers,following,tweet_count,listed_count,ca... | {'class_weight': 'balanced', 'criterion': 'entropy', 'n_estimators': 10} |
| 97786 | 17 | 0 | 28 | 32 | followers,following,tweet_count,listed_count,ca... | {'class_weight': 'balanced', 'criterion': 'entropy', 'n_estimators': 10} |
| 160709 | 17 | 0 | 28 | 32 | followers,following,tweet_count,listed_count,ca... | {'class_weight': 'balanced_subsample', 'criterion': 'gini', 'n_estimators': 10} |
| 163229 | 17 | 0 | 28 | 32 | followers,following,tweet_count,listed_count,ca... | {'class_weight': 'balanced_subsample', 'criterion': 'gini', 'n_estimators': 10} |
| 193476 | 17 | 0 | 28 | 32 | followers,following,tweet_count,listed_count,ca... | {'class_weight': 'balanced', 'criterion': 'gini', 'n_estimators': 10} |
| 195996 | 17 | 0 | 28 | 32 | followers,following,tweet_count,listed_count,ca... | {'class_weight': 'balanced', 'criterion': 'gini', 'n_estimators': 10} |
| 196059 | 17 | 0 | 29 | 31 | followers,following,tweet_count,listed_count,ca... | {'class_weight': 'balanced', 'criterion': 'gini', 'n_estimators': 10} |
| 196060 | 17 | 0 | 29 | 31 | followers,following,tweet_count,listed_count,ca... | {'class_weight': 'balanced', 'criterion': 'gini', 'n_estimators': 10} |
| 163292 | 17 | 0 | 29 | 31 | followers,following,tweet_count,listed_count,ca... | {'class_weight': 'balanced_subsample', 'criterion': 'gini', 'n_estimators': 10} |
| 163293 | 17 | 0 | 29 | 31 | followers,following,tweet_count,listed_count,ca... | {'class_weight': 'balanced_subsample', 'criterion': 'gini', 'n_estimators': 10} |
| 97849 | 17 | 0 | 29 | 31 | followers,following,tweet_count,listed_count,ca... | {'class_weight': 'balanced', 'criterion': 'entropy', 'n_estimators': 10} |
| 97850 | 17 | 0 | 29 | 31 | followers,following,tweet_count,listed_count,ca... | {'class_weight': 'balanced', 'criterion': 'entropy', 'n_estimators': 10} |

**Fig 7. Extract from resultsCR table, Costa Rica case.**

| num | TN | FP | FN | TP | cols | best |
|---|---|---|---|---|---|---|
| 65437 | 16 | 1 | 43 | 17 | followers,following,tweet_count,listed_count,ca... | {'class_weight': 'balanced_subsample', 'criterion': 'entropy', 'n_estimators': 10} |
| 196414 | 16 | 1 | 43 | 17 | followers,following,tweet_count,listed_count,ca... | {'class_weight': 'balanced', 'criterion': 'gini', 'n_estimators': 10} |
| 163647 | 16 | 1 | 43 | 17 | followers,following,tweet_count,listed_count,ca... | {'class_weight': 'balanced_subsample', 'criterion': 'gini', 'n_estimators': 10} |
| 98204 | 16 | 1 | 43 | 17 | followers,following,tweet_count,listed_count,ca... | {'class_weight': 'balanced', 'criterion': 'entropy', 'n_estimators': 10} |
| 98205 | 16 | 1 | 46 | 14 | followers,following,tweet_count,listed_count,ca... | {'class_weight': 'balanced', 'criterion': 'entropy', 'n_estimators': 10} |
| 163648 | 16 | 1 | 46 | 14 | followers,following,tweet_count,listed_count,ca... | {'class_weight': 'balanced_subsample', 'criterion': 'gini', 'n_estimators': 10} |
| 196415 | 16 | 1 | 46 | 14 | followers,following,tweet_count,listed_count,ca... | {'class_weight': 'balanced', 'criterion': 'gini', 'n_estimators': 10} |
| 65438 | 16 | 1 | 46 | 14 | followers,following,tweet_count,listed_count,ca... | {'class_weight': 'balanced_subsample', 'criterion': 'entropy', 'n_estimators': 10} |

**Fig 8. Extract from resultsNI table, Nicaragua case.**

**Table 4. Characteristics used in the models FP = 0 (Panama and Costa Rica) and FP = 1 (Nicaragua).**

| Panamá | rt_de_In, rp_de_in, rq_a_In, rq_de_In |
|---|---|
| Costa Rica | rt_a_In, rt_de_In, rp_a_In |
| Nicaragua | rq_a_In, rq_de_In |

characteristics. A pattern that can be observed at a glance is that the features that provided the best results in the three countries are combinations with retweets or retweets of messages (quotes).

## 2.6 Automatic labeling

From the first row of the results shown in Figs 6–8, the parameters and characteristics stored in those rows were used to generate the random tree models that classified the users (Users-Country_D2 tables) of each country. The results of the automatic classification were stored in the tables: classifiedUsersPA_D2, classifiedUsersCRA_D2 and classifiedUsersNI_D2, which will be mentioned as classifiedUsersCountry_D2, part of the results shown in Fig 9. These results included the columns: prob0, prob1, and pred. The prob0 column shows the probability of being classified as non-national (negative, class 0), the prob1 column shows the probability of being classified as national (positive, class 1), and the pred column shows the prediction decision, its value being 1 for national (positive) or 0 for a non-national user (negative). The decision threshold for classifying the user in class 1 was >0.5.

The same labeling procedure was also applied to Costa Rica and Nicaragua.

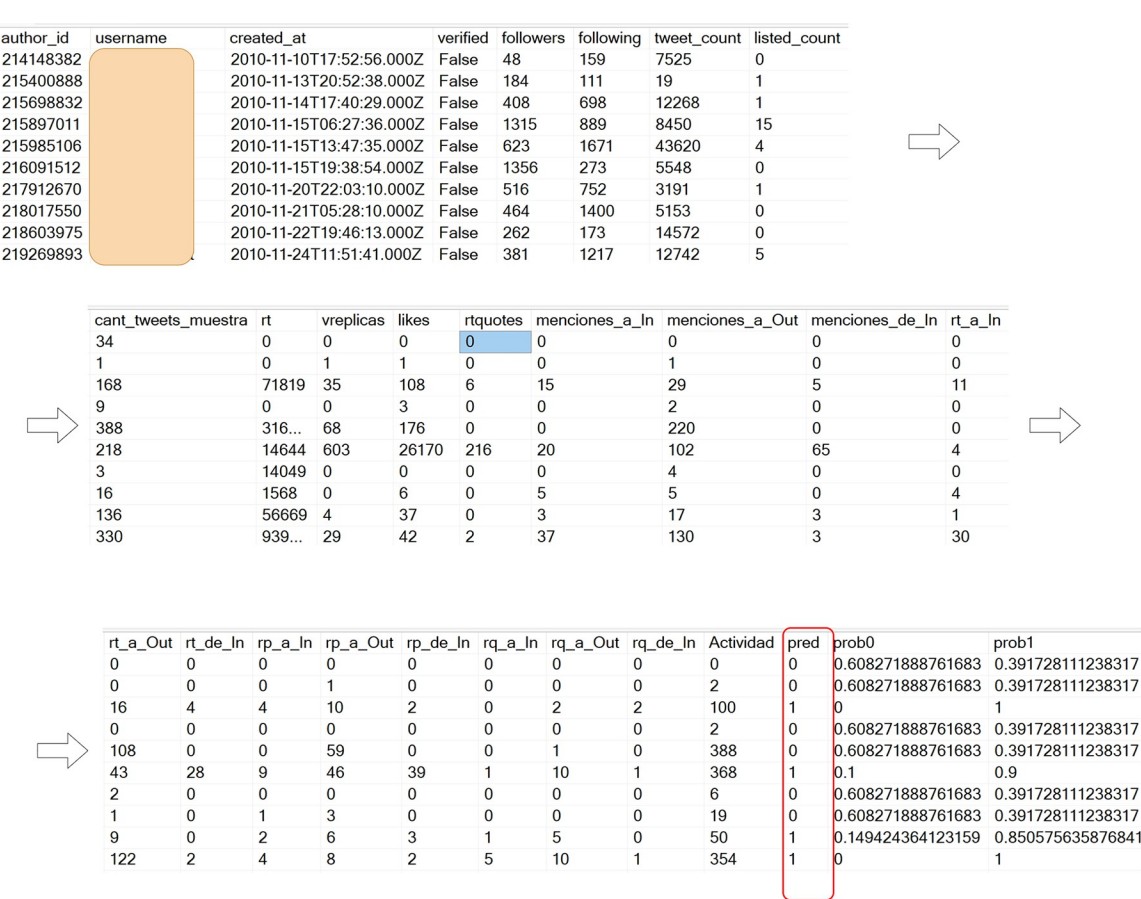

**Fig 9. An example of the results of the automatic labeling of users (classifiedUsersPA_D2 table).**

## 3. Results and discussion

The users that were classified 1 were then extracted from the classifiedUsersCountry_D2 tables and stored respectively in the tables: UsersPA_class1, UsersCR_class1 table and UsersNI_-class1; which will be mentioned as UsersCountry_class1.

To measure the accuracy of the model based on the classification results, random sampling was carried out for each UsersCountry_class1 (random sample confidence level of 95% and a sampling error of 5%). Each label was checked against the actual nationality of the samples to calculate the proportion of true positives.

Fig 10 shows the results comparing the percentage of nationals in the user samples from the UsersCountry_D2 tables (79.48%, 76.88% and 75.84%, average 77.40%) with the percentage of nationals in the user samples from the UsersCountry_class1 tables (94.03%, 91.17% and 89.61%, average 91.60%) that were automatically labeled.

When averaging the proportions inferred from the national samples in each country, the average is 77.4%, compared to 91.6% averaged after applying the automatic classification model, an average increase of 14.2%.Thus, there is a significant increase in the proportion of national users in the samples of automatically tagged users in the three countries. From 14,789 initial users in Panama, from which 79.48% of national users were estimated, a sample of 6,392 users was extracted with an estimated proportion of 94.03% of national users. In the case of Costa Rica, from 9,843 initial users and an estimated proportion of national users of 76.88%,

| User Tables | Users | Sample | Class 1 | Class 0 | % Class1 |
|---|---|---|---|---|---|
| Panamá (UsersPA_D2 table) | 14789 | 385 | 306 | 79 | 79.48% |
| Costa Rica (UsersCR_D2 table) | 9843 | 385 | 296 | 89 | 76.88% |
| Nicaragua (UsersNI_D2 table) | 5223 | 385 | 292 | 93 | 75.84% |

| Automatically tagged class 1 user tables | Users | Sample | Class 1 | Class 0 | % Class1 ( True Positive) |
|---|---|---|---|---|---|
| Panamá (UsersPA_class1 table) | 6392 | 385 | 362 | 23 | 94.03% |
| Costa Rica (UsersCR_class1 table) | 3886 | 385 | 351 | 34 | 91.17% |
| Nicaragua (UsersNI_class1 table) | 1343 | 385 | 345 | 40 | 89.61% |

**Fig 10. Comparative table of national proportions between the initial sample and the classified sample.**

3,886 users were extracted with an estimated proportion of national users of 91.17%. Finally, in the case of Nicaragua, with 5,223 initial users and an estimated proportion of national users of 77.40%, 1,343 users were extracted with an estimated proportion of national users of 89.61%. Although the number of users decreases when using the method in this study, the estimate of the proportion of national users increases considerably.

The most influential features in the classification model in the three countries studied were retweets and quotes. Exclusively numerical features were used in the automatic classification model. The fact that the model's features do not depend on textual content makes it generalizable to any other language, the only difficulty being accurately identifying the nationality of the users when labeling a small sample.

However, some biases and limitations should be taken into account when replicating this study. The greater the national diversity in a country, the more difficult it is to separate the nationality of origin from the nationality of residence. This is the case of the USA, which contains a great diversity of nationalities as a result of the construction of the state from the European emigration to America and subsequent waves of emigration from different geographies of the planet; therefore, there are people from the country expressing their messages in different languages and showing habits of different cultures. At the same time, there are Countries with a low level of national expression in the tweets that complicates their identification of the Country trait, unlike the Countries considered in this study whose users frequently express their nationality.

The study identified groups of users who intentionally avoid showing their nationalities, such as fan groups, users who promote their sexual services, and users who are part of international or regional organizations.

Depending on the above, the expressions are likely to increase or decrease when they are identified in the first download.

In the study, there are few accounts with high popularity in the database of users. It has been concluded that the classification model identifies such accounts in the non-national class. This is because many popular accounts happen to be celebrities who visit the country; therefore, the model associates that the accounts with a high number of followers are not national.

On the other hand, the study is not segmented by provinces; therefore, it is possible that the accounts are concentrated in a territorial part of the country, for example, urban centers, and it is also possible that they are concentrated in a certain profile, for example, male, professional, etc.

The language was a selection criterion applied at the time of downloading the Twitter messages. It is clear that this criterion cannot be applied to Countries with a great diversity of languages or different from Spanish one.

## 4. Conclusion

The proposed method was able to group samples of national users of important sizes for opinion or sentiment studies with an estimated proportion of users with country traits above 90% in the three Central American countries. The smallest sample size corresponds to Nicaragua, with 1,343 users, and the largest one is 6,392 users (Panama), which are important quantities for opinion studies. The use of Random Forests to generate classification models with small training samples, and the use of exclusively numerical characteristics based on the number of times that different interactions occur between users in the study sample, reaching the following results being correct and satisfactory. It was found that the most influential features in the classification model in the three countries studied were retweets and quotes.

A significant result was to test the validity of using exclusively numerical classification model features (non textual considerations). Consequently, the model's features do not depend on textual content, making it as generalizable to any other language such as English one. Experimental evaluation with real data demonstrates the efficiency and effectiveness of the method proposed by this study.

Finally, future studies are suggested to expand the coverage of Spanish-American countries to check if the pattern of increasing estimation of the proportion of nationals is maintained and if the method presented in this study is a generalizable route.

## Author Contributions

**Conceptualization:** Damián Quijano, Richard Gil-Herrera.

**Data curation:** Damián Quijano.

**Formal analysis:** Damián Quijano.

**Funding acquisition:** Richard Gil-Herrera.

**Investigation:** Damián Quijano, Richard Gil-Herrera.

**Methodology:** Damián Quijano.

**Resources:** Damián Quijano.

**Software:** Damián Quijano.

**Supervision:** Richard Gil-Herrera.

**Validation:** Damián Quijano, Richard Gil-Herrera.

**Visualization:** Damián Quijano.

**Writing – original draft:** Damián Quijano.

**Writing – review & editing:** Richard Gil-Herrera.

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
