## [Editor Report · Decision Letter 0]

16 Nov 2022

PONE-D-22-30387Methodological proposal to identify the nationality of Twitter users through Random-ForestsPLOS ONE

Dear Dr. Quijano Abad,

Thank you for submitting your manuscript to PLOS ONE. After careful consideration, we feel that it has merit but does not fully meet PLOS ONE’s publication criteria as it currently stands. Therefore, we invite you to submit a revised version of the manuscript that addresses the points raised during the review process.

Check the figure notations and numbering carefully.

We look forward to receiving your revised manuscript.

Kind regards,

Sathishkumar V E

Academic Editor

PLOS ONE 

Journal Requirements:

"Funded studies

Initials of the authors who received each award: DQ , RG

Grant numbers awarded to each author: 902.50$, 902.50$

The full name of each funder:

Universidad Especializada de las Américas

Universidad Internacional de la Rioja

URL of each funder website:

udelas.ac.pa

" ext-link-type="uri" xlink:type="simple">https://www.unir.net/"

---

## [Author Response · Author response to Decision Letter 0]

7 Dec 2022

1. Observation 1. Ensure that your manuscript meets PLOS ONE's style requirements, including those for file naming.

-The following changes were made to the manuscript: 

a. The author's section was completely modified, it was adjusted to what is indicated in formatting sample title authors

b. Keywords were removed

c. Increase space between Abstrac and first line

 Increase space between Related Work and the first line

d. Fig. 4. to Fig 4. : adjusts to what is indicated in format sample main body

 Fig. 5. to Fig 5. : :adjusts to what is indicated in format sample main body

e. Fig 7. Extract from resultsNI table, Nicaragua case it is changed to Fig 8. Extract from resultsNI table , Nicaragua case. 

The figure is the same, the error was the number.

2. Observation 2. We note that the grant information you provided in the ‘Funding Information’ and ‘Financial Disclosure’ sections do not match.

-You are right. We are wrong. We are confused, we have not received financial support in our research. 

In the ‘Funding Information’ section.We will write that we have not received funds for research. 

The Financial Disclosure section does not appear during the review submission, I cannot correct the data.

I have corrected:Unfunded studies: The author(s) received no specific funding for this work.

3. Observation 3. About financial disclosure.

The Financial Disclosure section does not appear during the review submission, I cannot correct the data.

I have corrected:Unfunded studies: The author(s) received no specific funding for this work.

4. Observation 4. About Data Availability statement, you have not specified.

-The study data mainly used 30 tables stored in two repositories:

Repository 1 has 12 tables: 

https://github.com/damian-quijano/identify_nationality_twitter

http://dx.doi.org/10.5281/zenodo.7254532

Repository 2 has 18 tables: 

https://github.com/damian-quijano/-identify_nationality_twitter2

http://dx.doi.org/10.5281/zenodo.7254534

This information was included in the submit (first time) in the section Additional information- Data Availability

It is fully accessible. All the tables and data mentioned in the manuscript are downloadable from the repositories.

This information, about full access to the data, is mentioned in section 2.2 Data collection of the manuscript.

We have confirmed that the repository data appears in the submit (the previous and the current one).

---

## [Editor Report · Decision Letter 1]

22 Dec 2022

Methodological proposal to identify the nationality of Twitter users through Random-Forests

PONE-D-22-30387R1

Dear Dr. Quijano Abad,

We’re pleased to inform you that your manuscript has been judged scientifically suitable for publication and will be formally accepted for publication once it meets all outstanding technical requirements.

Kind regards,

Sathishkumar V E

Academic Editor

PLOS ONE

Additional Editor Comments (optional):

Reviewers' comments:

quillbot-extension-portal/quillbot-extension-portal

---

## [Editor Report · Acceptance letter]

17 Jan 2023

PONE-D-22-30387R1 

Methodological proposal to identify the nationality of Twitter users through Random-Forests 

Dear Dr. Quijano:

I'm pleased to inform you that your manuscript has been deemed suitable for publication in PLOS ONE. Congratulations! Your manuscript is now with our production department. 

Kind regards, 

on behalf of

Dr. Sathishkumar V E 

Academic Editor

PLOS ONE